

# The *Argyreia collinsiae* species complex (Convolvulaceae): phenetic analysis and geographic distribution reveal subspecies new to science

Poompat Srisombat[1,2,3,*], Natthaphong Chitchak[1,*], Pantamith Rattanakrajang[4], Alyssa B. Stewart[1] and Paweena Traiperm[1]

[1] Department of Plant Science, Faculty of Science, Mahidol University, Bangkok, Thailand
[2] Department of Pharmaceutical Botany, Faculty of Pharmacy, Mahidol University, Bangkok, Thailand
[3] M.Sc. Program in Plant Science, Faculty of Graduate Studies, Mahidol University, Nakhon Pathom, Thailand
[4] Department of Pharmacognosy and Pharmaceutical Botany, Faculty of Pharmaceutical Sciences, Chulalongkorn University, Bangkok, Thailand
[*] These authors contributed equally to this work.

Corresponding author
Paweena Traiperm,
paweena.tra@mahidol.edu

## ABSTRACT

*Argyreia* Lour. is a speciose genus in the Convolvulaceae. However, the genus contains several problematic species complexes due to their morphological similarity. In this study, we aimed to resolve the *Argyreia collinsiae* complex, which consists of four similar operational taxonomic units (OTUs), *i.e.*, *A. collinsiae* (Craib) Na Songkhla & Traiperm, *A. dokmaihom* Traiperm & Staples, *A. versicolor* (Kerr) Staples & Traiperm, and a peculiar OTU typically known as the large-bract morphotype of *A. collinsiae*. Following morphological comparison and phenetic analysis, all four OTUs were found to be distinct. However, the large-bract morphotype of *A. collinsiae* was confirmed to be more closely related to the original morphotype of *A. collinsiae* than to the other two species. Species distribution modeling (SDM) was then conducted for both morphotypes of *A. collinsiae*, revealing different geographical ranges of suitable habitat for each. In conclusion, the large-bract morphotype of *A. collinsiae* was described in this study as a new subspecies, *A. collinsiae* subsp. *megabracteata* Traiperm & Srisombat, subsp. nov., based on morphological differences and separate geographic range. We also provide here an identification key, description, detailed illustrations, distribution data, and ecological notes of the new subspecies. An updated description of *A. versicolor* and an assessment of its conservation status were also prepared since the original description was based solely on dried herbarium specimens and lacks key details.

## INTRODUCTION

*Argyreia* Lour. is a large genus in the Convolvulaceae, currently comprising more than 143 species (*Shalini, Lakshminarasimhan & Maity, 2017*; *Staples & Traiperm, 2017*; *Chitchak et al., 2018*; *Traiperm et al., 2019*; *Lawand & Shimpale, 2020*; *Traiperm & Suddee, 2020*; *Rattanakrajang et al., 2022*; *Zhang, He & Liu, 2023*). It is primarily distributed in tropical

Asia, especially in seasonal dry climate zones (*Staples & Traiperm, 2017*). Thailand has been regarded as a center of species richness for this genus because about one-third of all *Argyreia* species inhabit this country (*Staples, Traiperm & Chow, 2015*). The extensive, ongoing floristic surveys in the country sometimes lead to the discovery of unknown *Argyreia* taxa, some of which have resulted in problematic species complexes caused mainly by morphological similarity to existing species. To date, many cases have been resolved either by traditional taxonomic practices, numerical methods, phylogenetic methods, or combined approaches (*Traiperm & Staples, 2014*; *Traiperm & Staples, 2016*; *Staples, Traiperm & Chow, 2015*; *Chitchak et al., 2018*; *Rattanakrajang, Traiperm & Staples, 2018*; *Traiperm & Suddee, 2020*; *Rattanakrajang et al., 2022*).

The complex of *Argyreia collinsiae* (Craib) Na Songkhla & Traiperm comprises three *Argyreia* species that have campanulate corollas, lanceolate to ovate bracts, and cordate leaves. The taxa within this complex are even more difficult to differentiate when plant materials come in the form of pressed and dried herbarium specimens. This limitation often impedes *Argyreia* identification and often leads to confusion and misidentification (*Staples, Traiperm & Chow, 2015*). The species usually considered in this species complex, besides *A. collinsiae* itself (*Khunwasi et al., 2005*), include *A. versicolor* (Kerr) Staples & Traiperm and *A. dokmaihom* Traiperm & Staples. Our field surveys in the Eastern and Northeastern regions of Thailand revealed an uncertain taxon that resembled the appearance of *A. collinsiae* in most aspects, except for its bract characters, which were explicitly different and more similar to those of *A. versicolor*. Specifically, instead of having lanceolate or oblanceolate bracts that detach during bloom as described in the original publication of *A. collinsiae* (*Craib, 1916*), this peculiar plant possessed relatively larger ovate or obovate bracts similar to *A. versicolor,* that persist throughout the blooming period. Thus, we labelled it the "large-bract morphotype" of *A. collinsiae* (*vs.* the original *A. collinsiae* morphotype) until its taxonomic status could be confirmed. Moreover, geographically, the original morphotype seemed to be distributed nearer to the coast while the large-bract morphotype was found in more inland areas of the country.

In this study, therefore, we aimed to delimit the taxonomic status of the large-bract *A. collinsiae* morphotype by examining it against the original *A. collinsiae* morphotype, together with the two other taxa in the species complex, *A. versicolor* and *A. dokmaihom*. We performed phenetic analyses using a combination of macro- and micromorphological data, as they have been proven valuable in differentiating morphologically similar species within *Argyreia* (*Staples, Traiperm & Chow, 2015*; *Traiperm et al., 2017*; *Chitchak et al., 2018*). Additionally, a preliminary assessment of the potential distribution range for each *A. collinsiae* morphotype was conducted using species distribution modelling, and the results are discussed in conjunction with data from field observations and herbarium specimens. We propose the taxonomic status of the large-bract *A. collinsiae* morphotype at the subspecific level, *A. collinsiae* subsp. *megabracteata*, subsp. nov. Furthermore, we provide an updated description of *A. versicolor,* as well as an identification key to the *A. collinsiae* species complex.

# MATERIALS & METHODS

The electronic version of this article in Portable Document Format (PDF) will represent a published work according to the International Code of Nomenclature for algae, fungi, and plants (ICN), and hence the new names contained in the electronic version are effectively published under that Code from the electronic edition alone. In addition, new names contained in this work which have been issued with identifiers by IPNI will eventually be made available to the Global Names Index. The IPNI LSIDs can be resolved and the associated information viewed through any standard web browser by appending the LSID contained in this publication to the prefix "http://ipni.org/". The online version of this work is archived and available from the following digital repositories: PeerJ, PubMed Central SCIE, and CLOCKSS.

## Plant collection & specimen identification

Materials for morphological investigation and phenetic analysis employed 19 plant collector numbers (17 populations) representing four operational taxonomic units (OTUs) (*i.e.,* four groups of morphologically similar plants), including the large-bract *A. collinsiae* morphotype (eight collector numbers; seven populations), the original *A. collinsiae* morphotype (six collector numbers; six populations), *A. versicolor* (three collector numbers; two populations), and *A. dokmaihom* (two collector numbers; two populations). They were collected from natural habitats following the localities reported in the *Flora of Thailand* (*Staples & Traiperm, 2010*), and from additional reports, such as observations spotted by locals and on social media (Table 1, Fig. 1). Inflorescences containing opened flowers and mature leaves were collected from individual plants (one individual plant per collector number), and were fixed in 70% ethyl alcohol (v/v) for morphological examination. Voucher specimens were prepared according to conventional methods for plant taxonomy (*Davies, Drinkell & Utteridge, 2023*), and deposited at the Forest Herbarium, Thailand (BKF). Identification was conducted using the *Flora of Thailand* (*Staples & Traiperm, 2010*) and the *Flora of Cambodia, Laos and Vietnam* (*Staples, 2018*). Plant materials were compared with the original descriptions of *A. collinsiae*, *A. versicolor*, and *A. dokmaihom*, and with high-resolution digital images of the type specimens retrieved from online herbaria databases (K and BM herbaria).

Distribution maps of all four OTUs were created from the coordinates of plants listed in Table 1 and additional information from herbarium specimens kept in BCU, BK, BKF, BM and K herbaria (see details in "Additional specimens examined" under the "Taxonomic treatment" section). A digital elevation model (DEM) and a basin boundary map were made in R (*RCore Team, 2022*) using the package "raster" (*Hijmans, 2023*) and "tmap" (*Tennekes, 2018*). The Hydro Basin level 3 shapefile for the study area was obtained from the Open Development Mekong DataHub (https://opendevelopmentmekong.net/) (*Pagès, 2004*). Semi-automated assessment of statistical conservation was conducted using Kew's Geospatial Conservation Assessment Tool (GeoCAT, *Bachman et al., 2011*, https://geocat.iucnredlist.org) to obtain the Area of Occupancy (AOO) and Extent of Occurrence (EOO), and following IUCN protocol (*IUCN Standards and Petitions Committee, 2022*).

**Table 1** List of specimens used for phenetic analyses.

| OTUs | Locality | Voucher |
|---|---|---|
| *Argyreia collinsiae* "Original morphotype" | Samut Prakan: Mueang Samut Prakan district | *P. Rattanakrajang, N. Chitchak, P. Srisombat & P. Traiperm 143* |
| | Chonburi: Mueang Chonburi district | *P. Rattanakrajang, N. Chitchak, P. Srisombat & P. Traiperm 144* |
| | Rayong: Mueang Rayong district | *P. Rattanakrajang, N. Chitchak, P. Srisombat & P. Traiperm 146* |
| | Chonburi: Mueang Chonburi district | *Y. Sirichamorn (2018) 10* |
| | Saraburi: Mueang Saraburi district | *P. Traiperm, N. Chitchak, P. Rattanakrajang & P. Srisombat 624* |
| | Phetchaburi: Cha-am district | *P. Rattanakrajang, P. Srisombat, A. Jirabunjongkij & P. Traiperm 150* |
| *Argyreia collinsiae* "Large-bract morphotype" | Nong Khai: Tha Bo district | *P. Hassa 17* |
| | Maha Sarakham: Wapi Pathum district | *N. Chitchak & P. Traiperm 23* |
| | Maha Sarakham: Na Dun district | *N. Chitchak & P. Traiperm 24* |
| | Surin: Mueang Surin district | *N. Chitchak & P. Traiperm 25* |
| | Roi Et: Phon Thong district | *N. Chitchak & P. Traiperm 26* |
| | Nakhon Ratchasima: Chok Chai district | *P. Traiperm, T. Sando, P. Rattanakrajang, N. Chitchak & P. Srisombat 625* |
| | Nakhon Ratchasima: Chok Chai district | *P. Traiperm, T. Sando, P. Rattanakrajang, N. Chitchak & P. Srisombat 626* |
| | Nakhon Ratchasima: Chok Chai district | *P. Traiperm, T. Sando, P. Rattanakrajang, N. Chitchak & P. Srisombat 627* |
| *Argyreia versicolor* | Sa Kaeo: Watthana Nakhon district | *P. Traiperm, C. Rattamanee, P. Rattanakrajang, N. Chitchak & P. Srisombat 628* |
| | Sa Kaeo: Watthana Nakhon district | *P. Traiperm, C. Rattamanee, P. Rattanakrajang, N. Chitchak & P. Srisombat 629* |
| | Sa Kaeo: Watthana Nakhon district | *A. Jirabanjongjit 08* |
| *Argyreia dokmaihom* | Kanchanaburi: Sangkhlaburi district | *G. Staple, P. Traiperm, W. Inta & N. Triyutthachai 1546* |
| | Kanchanaburi: Thong Pha Phum district | *P. Rattanakrajang, N. Chitchak & P. Traiperm 137* |

## Morphological examination

Qualitative and quantitative characters of leaves and inflorescences were investigated and categorized as shown in Table S1, with the assistance of stereomicroscopy (SM; Olympus SZX7). Terminology for general morphological characters followed *Beentje (2016)*. Light microscopy (LM; Olympus CX21) was also used to observe microscopic details of the staminal trichomes found on the lower part of filaments following methods described in *Chitchak, Stewart & Traiperm (2024)*. Briefly, portions of the stamen base containing the entire distribution area of staminal trichomes were cut, cleared by alkaline solution and stained with toluidine blue O. Qualitative and quantitative characters were observed and measured using ToupView software. Microscopic terminology followed *Evert (2006)*.

## Statistical analyses

All statistical analyses were performed using R (*RCore Team, 2022*). Both qualitative and quantitative characters, as listed in Table S1, were used when performing a Factor

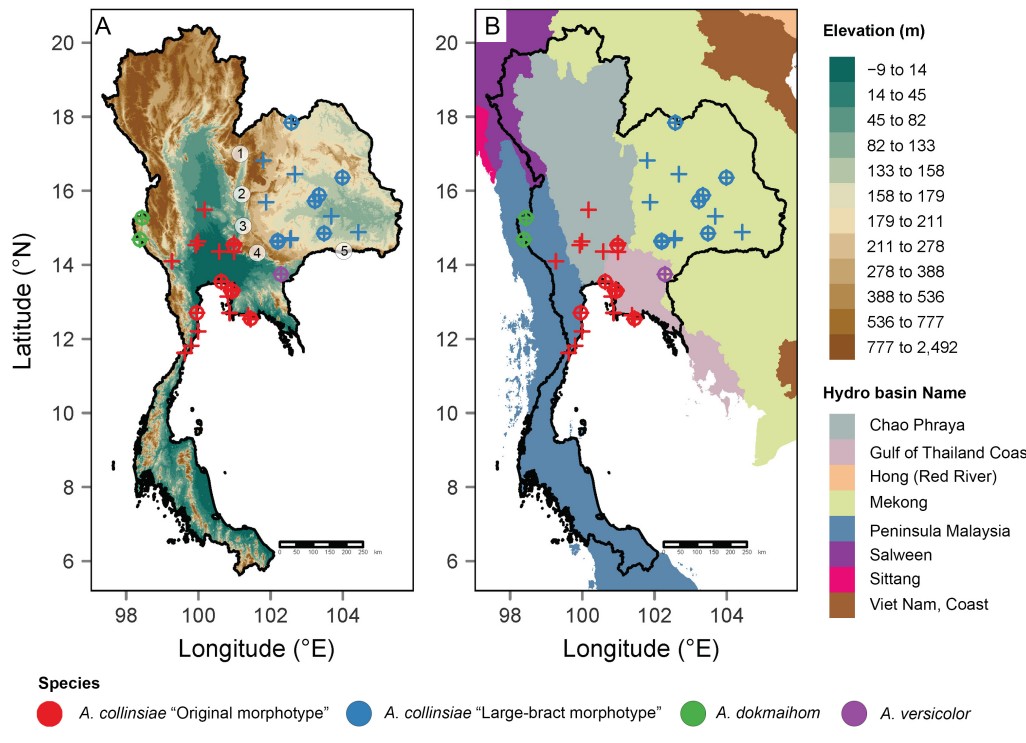

**Figure 1 Occurrence records of the original *Argyreia collinsiae* morphotype, a large-bract morphotype, *A. dokmaihom* and *A. versicolor* in Thailand.** Elevation map (A); basin boundary map (B). Circled numbers denote mountain ranges separating the two morphotypes: 1, Northern Phetchabun Range; 2, Southern Phetchabun Range; 3, Dong Phayayen Range; Sankamphaeng Range; 5, Phanom Dongrak Range. Plus signs (+) refer to occurrence data extracted from herbarium specimens and circles (o) refer to occurrence records of plant materials collected in this study. Digital elevation model (DEM) and basin boundary map were created by P. Srisombat using R (R Core Team, 2022) with the package "raster" (*Hijmans, 2023*) and "tmap" (*Tennekes, 2018*). Hydro Basin level 3 shapefile for the study area was obtained from the Open Development Mekong DataHub (https://data.opendevelopmentmekong.net/dataset/greater-mekong-subregions-hydro-basins-level-3).

Analysis of Mixed Data (FAMD) with the "FactoMineR" package (*Lê, Josse & Husson, 2008*). FAMD is a principal component method used to explore data comprising both continuous and categorical variables for investigating the similarities between OTUs (*Pagès, 2004*). Quantitative characters with high contributions towards the first two dimensions of FAMD were used to create boxplots. ANOVA and nonparametric Kruskal–Wallis tests were performed to test for significant differences among OTUs and, for significant characters, Paired Wilcoxon tests and Tukey's Honestly Significant Difference tests (HSD) were then used for post hoc analyses.

## Preliminary assessment of species distribution modelling of *A. collinsiae* morphotypes

The coordinate data used to create the distribution maps were also used for species distribution modelling (SDM) of both morphotypes of *Argyreia collinsiae*. We used bioclimatic variables at a 2.5′ resolution obtained from the WorldClim version 2.1

online data portal (https://worldclim.org) (*Fick & Hijmans, 2017*) for model calibration. Multicollinearity among the bioclimatic variables was tested for each morphotype of *A. collinsiae* in *R* with the "usdm" package (*Naimi et al., 2014*). A Pearson correlation coefficient less than 0.7 and a variance inflation factor (VIF) less than 10 were selected for model calibration (*Guisan, Thuiller & Zimmermann, 2017*). Additionally, a terrain roughness index from the ENVIREM online data portal (https://envirem.github.io) (*Title & Bemmels, 2018*) was added to the variables set.

We used the "disk" function to randomly generate 1,000 pseudoabsences for each individual model, with a 20-km minimum separation between a pseudoabsence's location and true presence records. Occurrence records were split with 70% used for training and 30% used for testing, with 10 replications. Eleven modeling algorithms were used to build an ensemble, which were artificial neural network (ANN) (*Ripley, 2007*), classification tree analysis (CTA) (*Breiman, 2017*), extreme gradient boosting (Xgboost) (*Chen & Guestrin, 2016*), flexible discriminant analysis (FDA) (*Hastie, Tibshirani & Buja, 1994*), generalized additive model (GAM) (*Hastie, 2017*), generalized boosted model (GBM) (*Friedman, 2001*), generalized linear model (GLM) (*McCullagh, 2019*), maximum entropy (MaxEnt) (*Phillips, Anderson & Schapire, 2006*), multivariate adaptive regression splines (MARS) (*Friedman & Roosen, 1995*), random forest (RF) (*Breiman, 2001*) and surface range envelop (SRE) (*Busby, 1991*). Ensemble models were created using weighted mean algorithms in "biomod2" (*Thuiller et al., 2023*) and only individual models with an AUC and TSS score above 0.7 were considered (Table S2). The weighted mean approach enables the exclusion of poorly performing algorithms, and incorporating only the best-performing individual models in an ensemble has been shown to improve the overall performance of the ensemble model (*Hao et al., 2019*).

## RESULTS

### Morphological study

Leaf shape exhibited three types as indicated by the ratio of leaf blade length to width, *i.e.,* ovate (equal or greater than 1.4 but smaller than 1.6), broadly ovate (equal or greater than 1.2 but smaller than 1.4), and orbicular (greater than 0.9 but smaller than 1.1) (Fig. 2). Orbicular leaves were found in both morphotypes of *A. collinsiae*, while broadly ovate leaves were found in large-bract *A. collinsiae*, *A. dokmaihom* and *A. versicolor*. Finally, ovate leaves were observed only in *A. dokmaihom*. Three of the four OTUs exhibited two types of leaf apex, namely acute and acuminate, while *A. dokmaihom* presented only one type, which was acuminate. A cordate leaf base was observed in all four OTUs. The adaxial surfaces were scabrous-strigose in *A. versicolor*, strigose in *A. dokmaihom,* and puberulent in the two morphotypes of *A. collinsiae*. Moreover, brown hairs on both leaf surfaces appeared only in *A. dokmaihom,* while hairs on the other OTUs were greyish in color.

Inflorescences were arranged as crowded cymes in *A. dokmaihom*, whereas flowers of the other OTUs were lax. Bracts in three OTUs (the large-bract *A. collinsiae* morphotype, *A. versicolor*, and *A. dokmaihom*) were persistent throughout the flowering season, while the original *A. collinsiae* morphotype was caducous (Fig. 2).

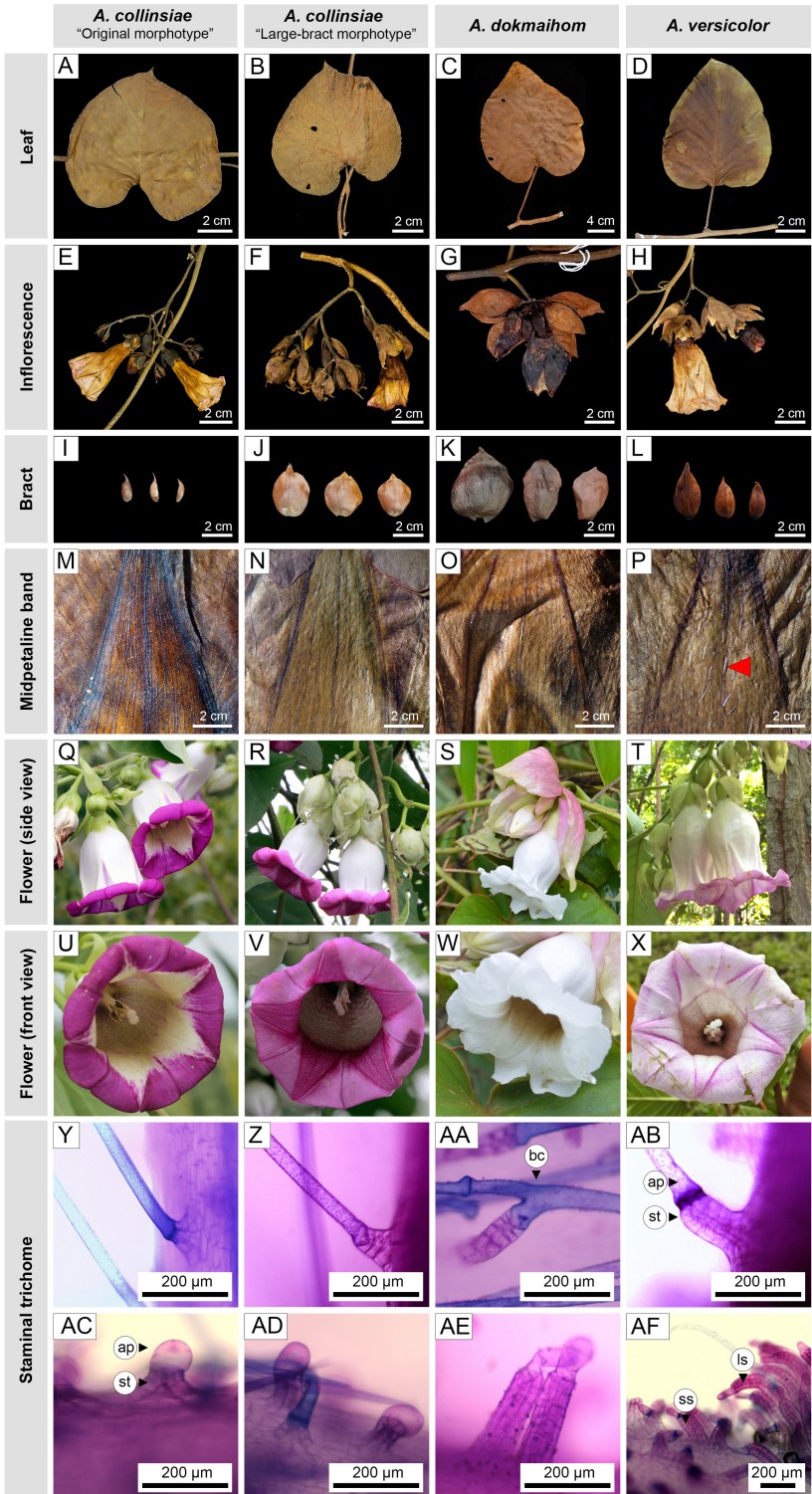

**Figure 2  Macro and micro morphology of the four studied *Argyreia* OTUs.** Original *A. collinsiae* morphotype (A, E, I, M: *Y. Sirichamorn (2018) 10*; (continued on next page...)

**Figure 2 (…continued)**
Q, U: *P. Rattanakrajang et al. 143*), large-bract *A. collinsiae* morphotype (B, F, J, N, R, V: *P. Hassa 17*), *A. dokmaihom* (C, G, K, O: *Staples et al. 1546*; S, W: *P. Rattanakrajang et al. 137*) and *A. versicolor* (D, H, L, P: *A. Jirabanjongjit et al. 08*; T, X: *P. Traiperm et al. 630*). Non-glandular trichomes (Y, Z, AA, AB) consisted of simple straight non-glandular trichomes (Y-Z, AB), and non-glandular trichomes with bicellular-branched apical cells (AA). Glandular trichomes (AC-AF) consisted of short-stalked glandular trichomes (AC-AF), and long-stalked glandular trichomes (AF). Photos by N. Chitchak (AA, AF), P. Hassa (R, V: *P. Hassa 17*), P. Rattanakrajang (S, T, W, X: *P. Rattanakrajang et al. 137, P. Traiperm et al. 630*), P. Srisombat (Y-Z, AB-AE) and W. Inta (Q, U: *P. Rattanakrajang et al. 143*). Red arrowhead indicates non-glandular trichome on the abaxial surface of the mid-petaline band. Abbreviations: ap, apical cell; bc, bicellular-branched apical cell; ls, long-stalked glandular trichome; ss, short-stalked glandular trichome; st, stalk.

All OTUs exhibited a pendant floral orientation with campanulate flowers. The sepal abaxial surface of the large-bract *A. collinsiae* morphotype and *A. dokmaihom* were glabrous. Additionally, the abaxial surface of the outer sepals of the original *A. collinsiae* morphotype were glabrous or rarely glabrescent, while the outer and middle sepals of *A. versicolor* were covered by hairs. The corolla tubes of all OTUs were white, but the limbs showed more variation in shade, consisting of purple, pale purple, and white. Purple limbs were observed in both morphotypes of *A. collinsiae*, while *A. versicolor* exhibited pale purple limbs. In contrast, the corolla limbs of *A. dokmaihom* were completely white (Fig. 2). The abaxial surface of the midpetaline bands of *A. versicolor* were covered by sparsely pilose hairs, whereas those of the other OTUs were glabrous. Stamens and pistils of all four OTUs were included within the corolla tube. Stamens of all OTUs were covered by trichomes along the lower part of the filaments. Glandular and non-glandular trichomes were present in all OTUs (Figs. 2Y–2Z, 2AA–2AF). Glandular trichomes were divided into two types according to the length of the stalk: short (123.14 ± 37.94 mm) and long (443.97 ± 101.75 mm) trichomes (Figs. 2AC–2AF). Non-glandular trichomes also exhibited two types: simple straight (Figs. 2Y–2Z, 2AB) and branched (Fig. 2AA). The short-stalked glandular trichomes occurred in both morphotypes of *A. collinsiae* and also in *A. dokmaihom*, while both short- and long-stalked glandular trichomes were present in *A. versicolor* (Fig. 2AF). The simple non-glandular trichomes were commonly observed in all OTUs (Figs. 2Y–2Z, 2AB), whereas the branched non-glandular trichomes were found only in *A. dokmaihom* (Fig. 2AA).

## Phenetic analysis

The first two dimensions of FAMD explained 29.9% and 19% of the overall variance. The results of FAMD showing the distribution of OTUs (based on study individuals) was plotted on the two-dimensional scatter plot (Fig. 3). Four discrete elliptic areas computed by t-distribution corresponded to *A. versicolor*, *A. dokmaihom*, and the two morphotypes of *A. collinsiae*. The contributions of each character toward the first two dimensions are visualised in Fig. S1. In total, there were 23 significant characters from the first-two dimensions. The 10 significant quantitative characters were outer bract width, 1st inner bract width, leaf blade length, outer bract length, 1st inner bract length, ratio of outer bract length to width, ratio of 1st inner bract length to width, leaf blade width, leaf basal extension length and non-glandular trichome stalk length. The 13 significant qualitative characters were adaxial

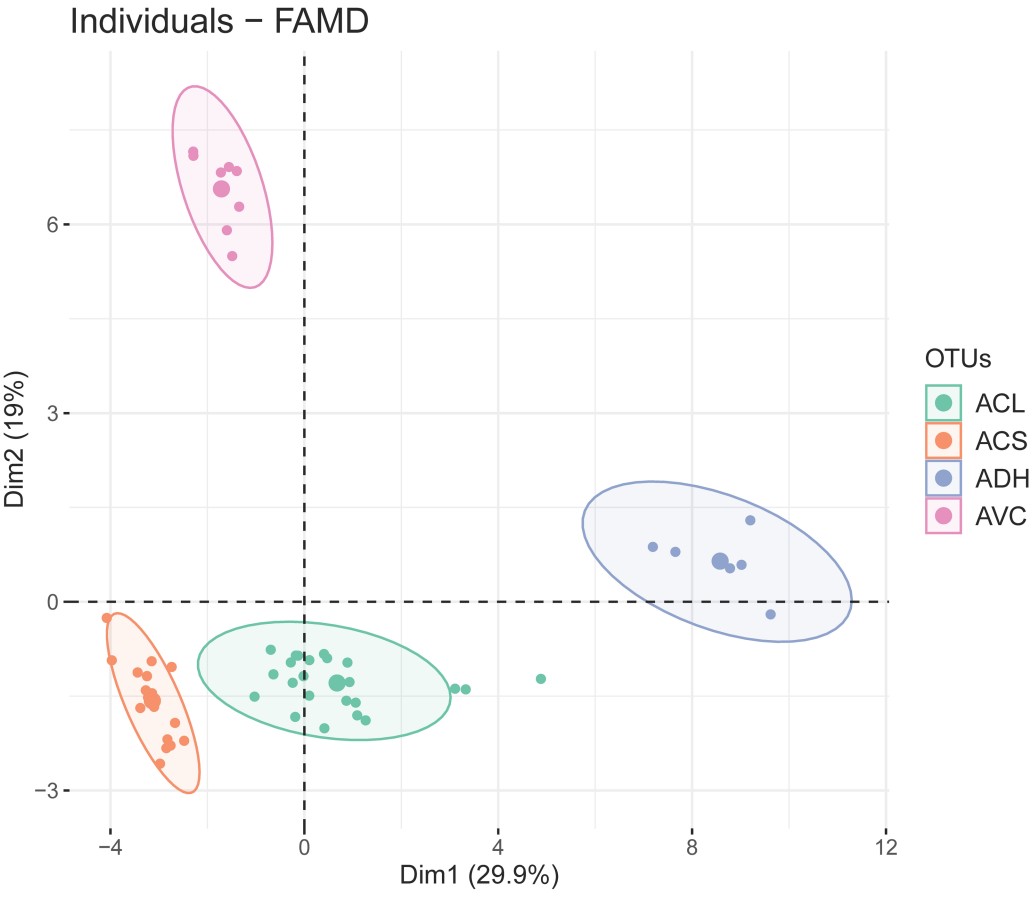

**Figure 3** **Two-dimensional scatter plot depicting the coordinates of each study individual in the FAMD (small dots) and the ellipse center (large dots).** Four distinct groups were generated by t-distribution, shown by different colors. ACL, Large-bract *A. collinsiae* morphotype; ACS, Original *A. collinsiae* morphotype; ADH, *A. dokmaihom*; AVC, *A. versicolor*.

leaf surface, corolla limb color, adaxial leaf surface color, inflorescence density, smell, presence of branched non-glandular trichomes on the base of stamens, leaf shape, bract persistence, presence of hairs on midpetaline bands, presence of long-stalked glandular trichomes on the base of stamens, presence of hairs on middle sepals, presence of hairs on outer sepals, and leaf apex. We utilised the first five dimensions, which accounted for 72.57% of the overall variation.

ANOVA and Kruskal-Wallis tests showed that 10 quantitative characters from FAMD analysis were important for the identification of each OTU. The results of the paired Wilcoxon tests and Tukey's HSD tests showed that some characters can be utilized to distinguish between OTUs, such as leaf blade length, leaf blade width, and outer bract length (Fig. S2).

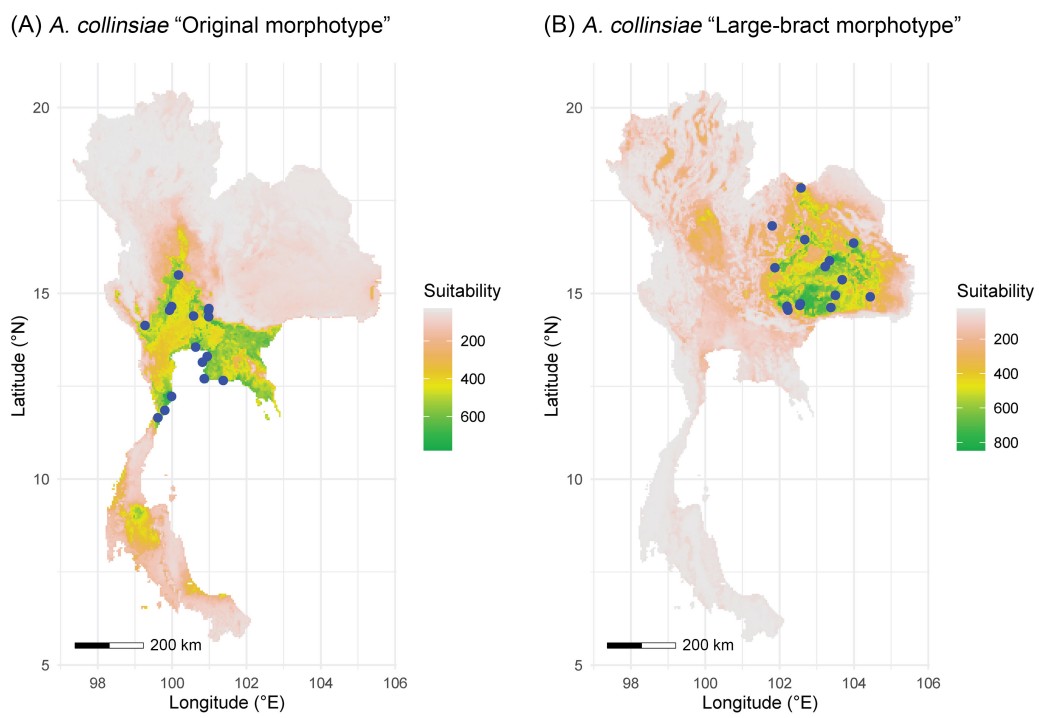

(A) *A. collinsiae* "Original morphotype"   (B) *A. collinsiae* "Large-bract morphotype"

**Figure 4   Areas of suitable habitat in Thailand for *Argyreia collinsiae*.** Original morphotype (A), large-bract morphotype (B); ensemble models combined with a weighted mean algorithm. Maps were created by P. Srisombat using R (*RCore Team, 2022*) with the "biomod2" package (*Thuiller et al., 2023*). Bioclimatic variables at a 2.5′ resolution was obtained from the WorldClim version 2.1 online data portal (https://worldclim.org) (*Fick & Hijmans, 2017*), and a terrain roughness index from the ENVIREM online data portal (https://envirem.github.io) (*Title & Bemmels, 2018*).

## Preliminary assessment of species distribution modelling of the two *A. collinsiae* morphotypes

The multicollinearity test showed different sets of bioclimatic variables for each morphotype of *A. collinsiae*. Four bioclimatic variables were selected for model calibration for the original *A. collinsiae* morphotype, while seven bioclimatic variables were used for the large-bract *A. collinsiae* morphotype (Table S3). The ensemble model performance for each morphotype of *A. collinsiae* was good, with TSS and AUC scores greater than 0.8 (Table S4). Variable importance scores showed that the minimum temperature of the coldest month (BIO6) contributed the most to the distribution of the original *A. collinsiae* morphotype, while terrain roughness index was the main determinant of the large-bract morphotype's distribution. The final ensemble model revealed that current suitable habitats for the original *A. collinsiae* morphotype are predominantly in the Southwestern, Central, and Southeastern regions of Thailand (Fig. 4A), while that of the large-bract *A. collinsiae* morphotype are in the Northeastern and Eastern regions of Thailand (Fig. 4B). Small patches of suitable habitat for the original *A. collinsiae* morphotype were also found in the Southern region, although there are no records of the species from this area.

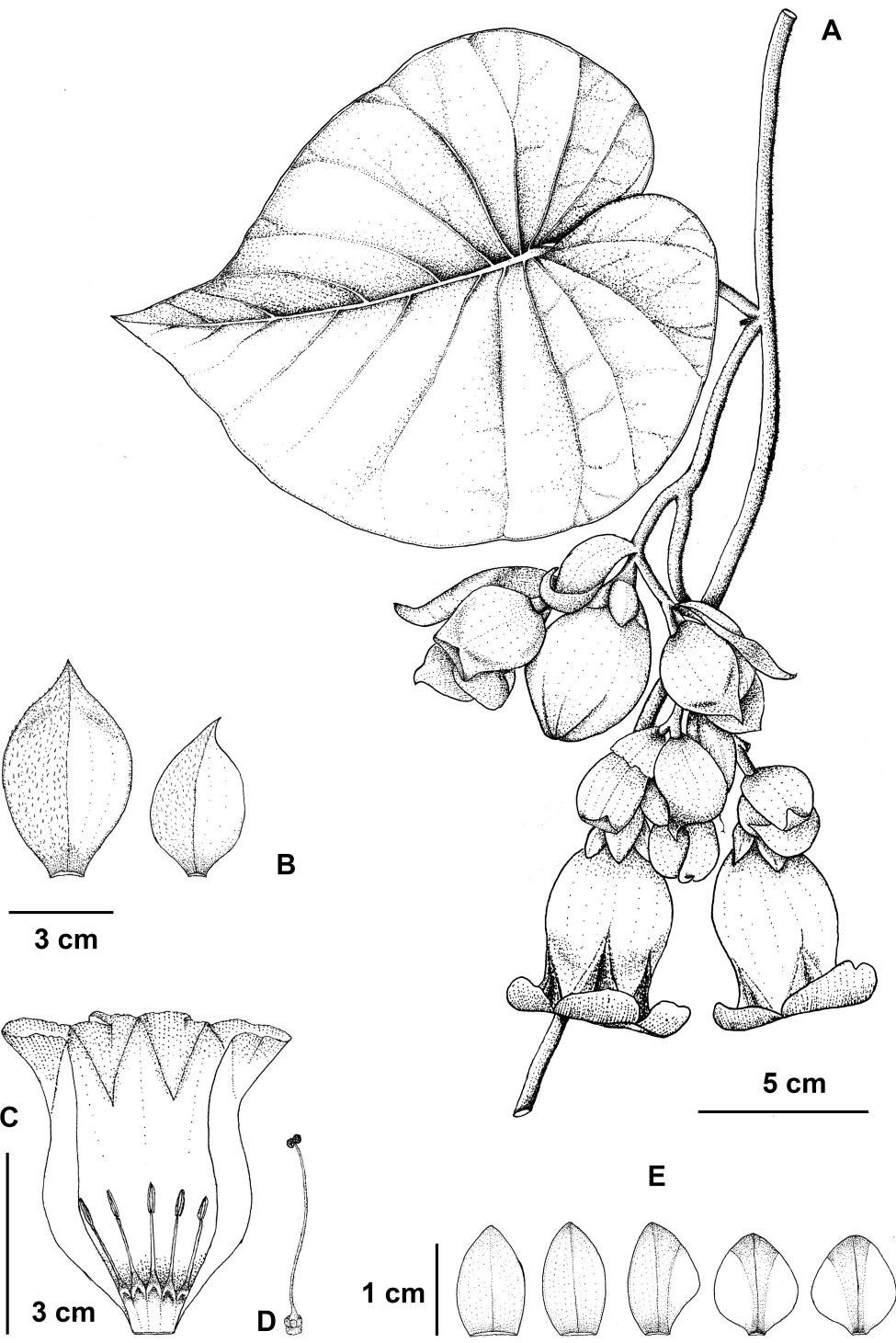

**Figure 5** *Argyreia collinsiae* **subsp.** *megabracteata* **Traiperm & Srisombat.** Flowering stem (A), bracts (B), opened corolla with stamens (C), pistil (D), and sepals, outer (left) to innermost (right) (E) drawn by P. Srisombat from *P. Traiperm et al. 625* and *627* (A-E).

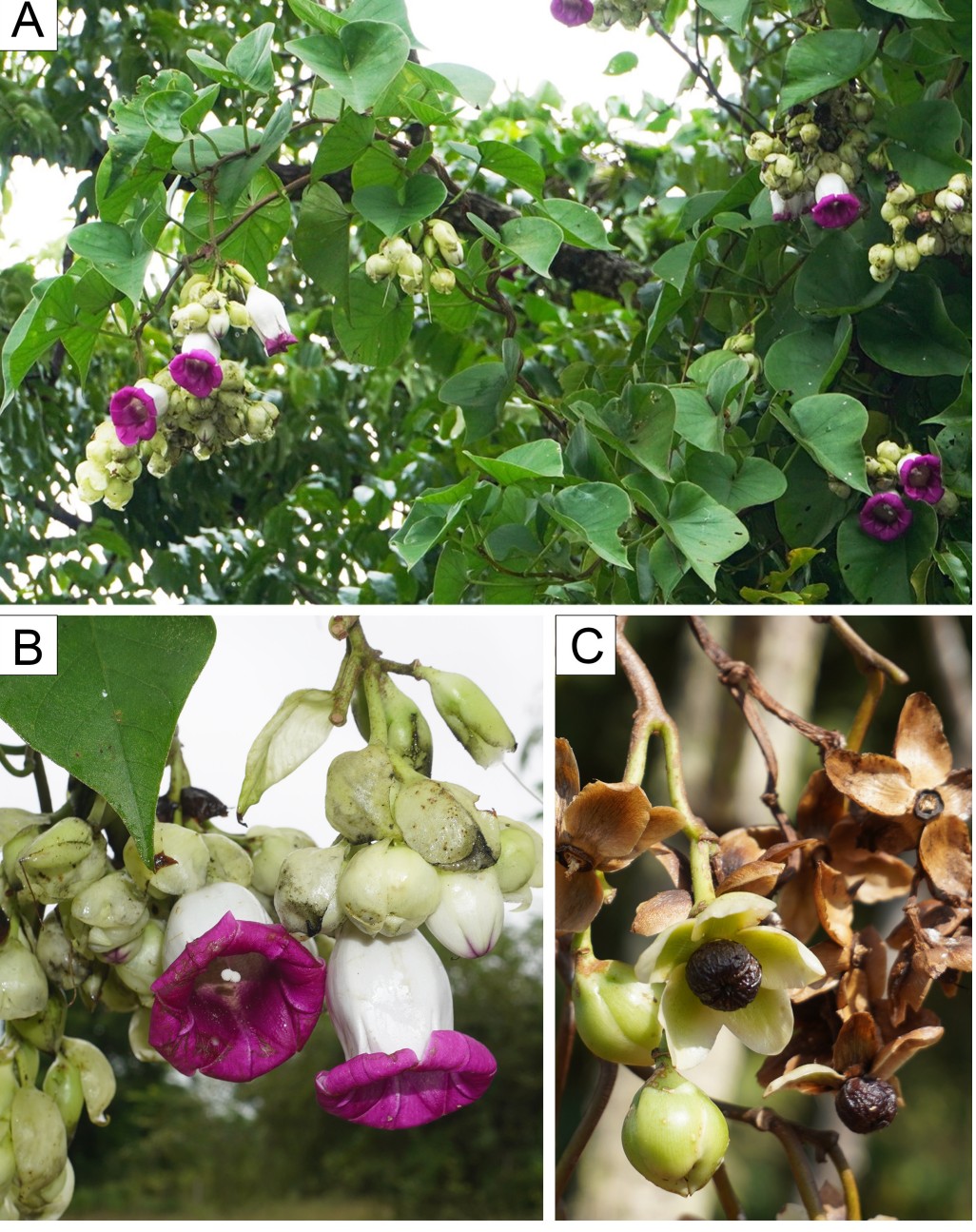

**Figure 6** *Argyreia collinsiae* **subsp.** *megabracteata* **Traiperm & Srisombat, subsp. nov.** Plant habit (A), inflorescence (B), and fruits (C). Photos by N. Chitchak; (A, B) *P. Traiperm et al. 625* and (C) *P. Srisombat 05*.

## DISCUSSION

One of the largest challenges for the Convolvulaceae is defining solid taxonomic boundaries. This difficulty stems in part from morphological lability caused by a mosaic pattern of evolution that has created blurred boundaries for several taxa, either at the generic, specific,

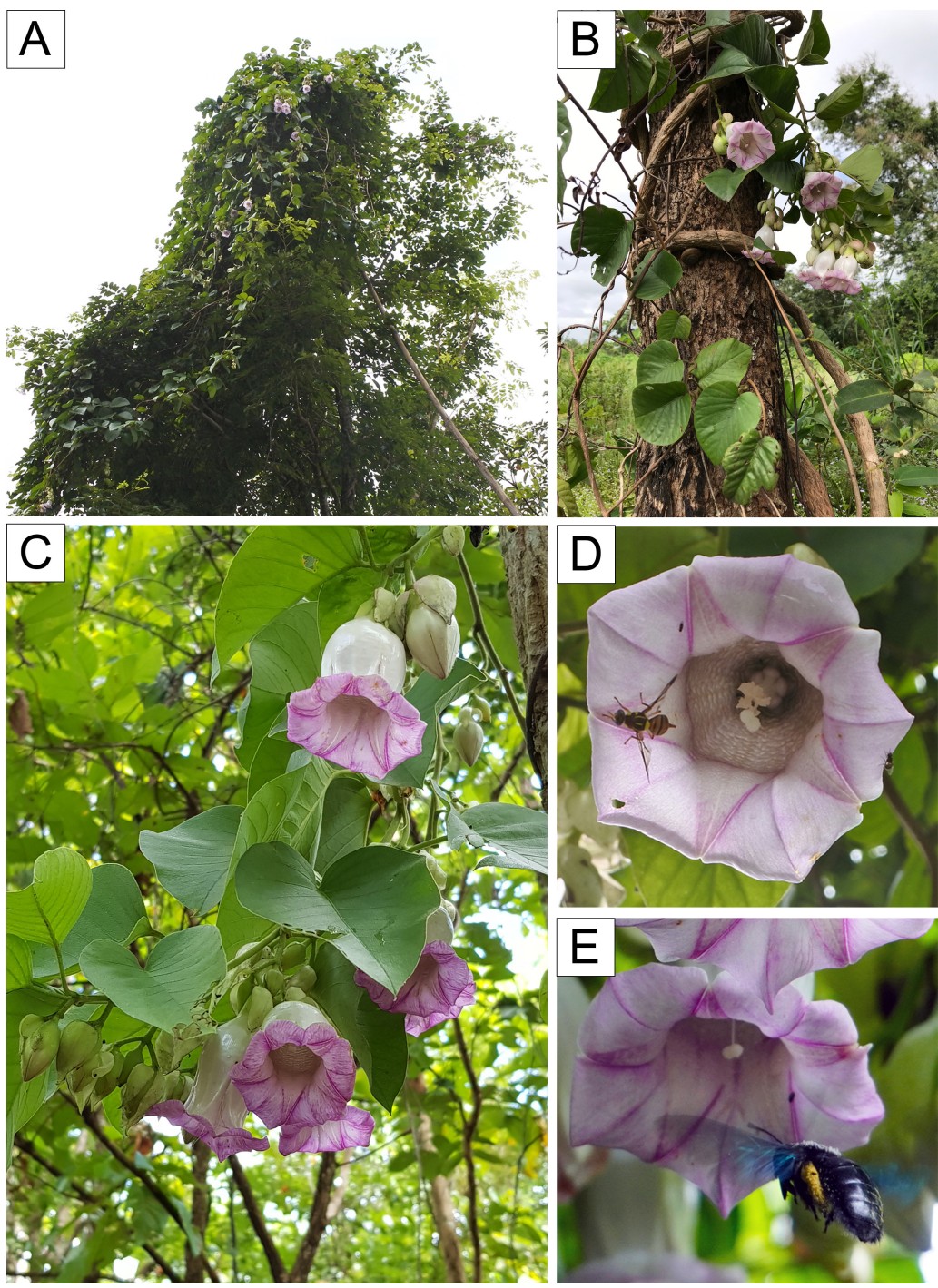

**Figure 7** ***Argyreia versicolor* (Kerr) Staples & Traiperm.** Plant habit (A–B), inflorescence (C), interaction with insect visitors, *Bactrocera* sp. and *Xylocopa* sp. (D–E). Photos by P. Rattanakrajang (A, D *P. Traiperm et al. 628*), P. Traiperm (B *P. Traiperm et al. 628*), Y. Sirichamorn (C *P. Traiperm et al. 628*) and N. Chitchak (E *Jirabanjongjit et al. 08*).

or infraspecific levels (*Manos, Miller & Wilkin, 2001*; *Simões & Staples, 2017*). Some species are highly polymorphic, such as *Ipomoea* spp. and *Daustinia montana* (*Santos, De Arruda & Buril, 2020*; *Alencar, Maciel & Buril, 2024*), while others may actually be complexes of morphologically similar taxa. Multiple species complexes have been untangled and their taxonomic circumscriptions clearly denoted, such as the *Cuscuta chinensis* complex (*Costea, Spence & Stefanović, 2011*), *Argyreia suddeeana* complex (*Traiperm et al., 2017*), *A. mekongensis* complex (*Chitchak et al., 2018*), and *Jacquemontia* complexes (*Belo et al., 2023*; *Belo, Louzada & Buril, 2024*). Most research examining species complexes in this family has been based on, and has proven the value of, macro- and micromorphological characters in combination with statistical and numerical approaches. Some complexes have also been resolved using molecular and genetic approaches (*e.g.*, *Carine & Robba, 2010*; *Costea, Spence & Stefanović, 2011*).

Our study conducted phenetic analysis using Factor Analysis of Mixed Data (FAMD) in order to delimit the taxonomic entity of a peculiar *Argyreia* sp. found within the *A. collinsiae* complex, which consists of *A. collinsiae, A. dokmaihom,* and *A. versicolor*. The results of this study corroborate the significance of bract characters for differentiation, as has also been shown in previous studies examining *A. suddeeana* (*Traiperm & Staples, 2014*), *A. albiflora* (*Staples, Traiperm & Chow, 2015*), and *A. gyrobracteata* (*Chitchak et al., 2018*).

The rediscovery of *A. versicolor* in 2018 (*Staples et al., 2021*), after its absence from the scientific community for almost a century following type specimen collection, did not only help resolve the taxonomic boundary of *A. collinsiae*, but also reinforces the status of *A. dokmaihom,* a species that was originally thought to be "*A. versicolor*" but was later described as a distinct species (*Traiperm & Staples, 2016*). Moreover, this rediscovery has also uncovered that some characteristics are not as described by (*Kerr, 1941*) and in the Convolvulaceae account of the *Flora of Thailand* (*Staples & Traiperm, 2010*), such as floral color and inflorescence orientation. These descriptions were made based on dried herbarium specimens, which had lost some characters that can only be seen in living plants.

The results of this study also show that, with careful observation, the members of the *A. collinsiae* species complex can be quickly told apart by corolla color: purple in *A. collinsiae* (although corolla color does not distinguish between its two morphotypes), pale purple in *A. versicolor*, and white in *A. dokmaihom*. But, herbarium specimens are easily confused, especially the large-bract *A. collinsiae* morphotype, which retains the corolla of the original *A. collinsiae* morphotype but displays bracts more similar to *A. versicolor.* This issue, where different taxa look similar when dry, has caused taxonomic problems within this family for a long time. Therefore, thoughtful field notes are essential. Moreover, some species in the Convolvulaceae exhibit intraspecific variation in corolla color, as found in *Argyreia variabilis* Traiperm & Staples (*Staples & Traiperm, 2008*) and *Ipomoea aquatica* (*Hassa, Traiperm & Stewart, 2020*). To improve the accuracy of identification, taxonomists should use corolla color in combination with other characters.

Apart from macromorphology, we found two important micromorphological traits for separating taxa within this complex, *i.e.,* midpetaline band hairs and staminal hairs. Indumentum on midpetaline bands is a common character used to circumscribe plants in

the Convolvulaceae, such as in *Argyreia* Lour. (*Staples & Traiperm, 2010*), *Bonamia* Thouars (*Wood, 2013*), *Distimake* Raf. (*Johnson, 2010*), *Duperreya* Gaudich. (*Johnson, 2009*), *Erycibe* Roxb. (*Syahida-Emiza, Staples & Haron, 2011*), *Ipomoea* L. (*Wood & Scotland, 2017*), and *Stictocardia* Hallier f. (*Johnson, 2004*). *Argyreia versicolor* is the only taxon in this species complex that possesses hairs on the abaxial midpetaline band. The taxonomic value of staminal hairs has long been acknowledged, especially in the discrimination of Malesian *Argyreia* (*van Ooststroom, 1950*; *van Ooststroom, 1952*; *van Ooststroom & Hoogland, 1953*), for clarifying members within the *A. mekongensis* complex (*i.e., A. mekongensis, A. gyrobracteata,* and *A. leucantha*; *Chitchak et al., 2018*), and in comparing the floral morphology of three sympatric *Argyreia* species (*Jirabanjongjit et al., 2024a*). The usefulness of staminal hairs found in the present study primarily stems from the presence of non-glandular apical cell branching (specific to *A. dokmaihom*) and the presence of long-stalked glandular trichomes (specific to *A. versicolor*), while the staminal hairs of both *A. collinsiae* morphotypes were homogeneous, suggesting the closer relation of the large-bract taxon to the original *A. collinsiae* morphotype rather than to *A. versicolor*.

Besides morphological differences, habitat and geographic differences are also worth mentioning. *Argyreia dokmaihom* is the only species restricted to the Southwestern region of the country, in mixed evergreen and deciduous forests, usually with bamboo, at 200 m asl (*Traiperm & Staples, 2016*). *Argyreia versicolor* is only known from its type locality in the Southeastern region. Two mature individuals were found growing in an open, disturbed plantation, at ca. 100 m asl. *Argyreia collinsiae* is the only widely distributed species (in Thailand, the distribution of the original morphotype spans 47,000 km$^2$ and that of the large-bract morphotype spans 72,000 km$^2$) and is found in diverse habitats, but usually in open, sunny areas, with most specimens collected from such habitats in Thailand (Fig. 1A). The data from this study suggest that the original and large-bract morphotypes have distinct geographic distributions. The original morphotype was formerly thought to be distributed near coastal areas (*Staples & Traiperm, 2010*), but, in fact, is also found hundreds of kilometers away from the coast. However, it seems to be restricted to lower elevations (3–94 m asl) within the Central region (Chao Phraya Basin), and neighboring areas to the north (Chao Phraya Basin), west (Malay peninsula) and east (Gulf of Thailand coast). In contrast, the large-bract morphotype is found only in the Mekong Basin at elevations ranging from 132–220 m asl in the Northeastern and Eastern regions of the country (Fig. 1). Five mountain ranges (*Choenkwan, Fox & Rambo, 2014*) may help serve as barriers separating the two morphotypes (*i.e.,* Northern Petchabun, Southern Petchabun, Dong Phayayen, Sankampaeng, and Phanom Dongrak ranges; Fig. 1A). Additionally, the suitable habitats for each morphotype of *A. collinsiae* predicted by species distribution modelling also correspond with our field observations, indicating that the two morphotypes potentially have different bioclimatic preferences. Specifically, terrain roughness appears to be an important determinant for the large-bract morphotype and minimum temperature during the coldest month (BIO6) restricts the distribution of the original morphotype.

Geographic isolation is thought to give rise to morphologically distinct populations, both of which are important criteria for infraspecific classification (*Hamilton & Reichard,*

*1992*). Additionally, data regarding genetics, reproductive isolation, and fertility further improve the decision-making process (*Stuessy, 2009*). In the Convolvulaceae, it is common practice for infraspecific classification to be based only on evidence of morphological and geographic differences, resulting in subsequent subspecies and varieties (*e.g., Carine & Robba, 2010*; *Johnson, 2012*; *Brummitt & Namoff, 2013*; *Mill, 2013*; *Wood & Scotland, 2017*). Subspecies in this family are typically located in populations in disjunct distribution areas, which can be thousands of kilometers away from other populations. Disjunct distributions, in some cases, can result from geographic barriers. For example, the Gulf of Eden separated *Convolvulus hystrix* subsp. *hystrix* Vahl in the Arabian Peninsula from *C. hystrix* subsp. *inermis* (Chiov.) J.R.I. Wood & R.W. Scotland and *C. hystrix* subsp. *ruspolii* (Dammer ex Hallier f.) J.R.I. Wood & R.W. Scotland in East Africa, while the latter two subspecies were separated from each other by a vast semi-arid landscape (*Wood et al., 2015*). In contrast, varietal classification is usually considered for distinct populations that may overlap within the distribution of the species or is sporadically distributed without an obvious geographic pattern (*Carine & Robba, 2010*; *Wood et al., 2015*).

Mountain ranges appear to be the main constraint separating the original and the large-bract morphotypes of *A. collinsiae*. In addition to climatic or habitat preference, there is possibly another factor underlying this distribution limitation, which is seed dispersal capability. The only unique characteristic that retains *Argyreia* as a segregate genus is its indehiscent berries, while its close relative, *Ipomoea,* possesses dehiscent capsules (*Stefanović, Austin & Olmstead, 2003*). Fruits of *Ipomoea* dehisce when mature and release seeds that are usually covered with hairs, which facilitates dispersal by wind or drifting along water surfaces (*Austin et al., 2001*; *Griz & Machado, 2001*; *Miryeganeh et al., 2014*). In contrast, *Argyreia* has evolved animal-mediated seed dispersal. The fruits of this genus are fleshy, pulpy, or mealy, usually with bright colors, such as red, orange, or yellow, which is probably an adaptation to dispersal by birds, specifically (*Staples & Traiperm, 2010*). In the case of *A. collinsiae,* the mealy, dark brown berries have a sweet taste (to human senses), suggesting that other frugivores, such as rodents or primates, may help disperse the seeds in addition to birds. It is noteworthy that one of our study populations grows sympatrically with the crab-eating macaque (*Macaca fascicularis* Raffles, 1821) (P Srisombat & N Chitchak, 2018, pers. obs.). It is possible that the non-overlapping distributions of the two morphotypes of *A. collinsiae* are actually due to limitations in movement or migration of seed dispersers, with a string of mountain ranges creating a geographic barrier between the two morphotypes.

The observed morphological differences (Table 2, Fig. 2), with support from phenetic analyses (Fig. 3) and the occurrence of geographic isolation (Figs. 1, 4), indicate that the large-bract morphotype of *A. collinsiae* should be treated as a novel subspecies, *A. collinsiae* subsp. *megabracteata*. We also provide an updated description of *A. versicolor* based on fresh materials, since the original description was based on observations of dried herbarium specimens and lacks key details (*Staples & Traiperm, 2010*). However, a full description of *A. dokmaihom* is not included here as it has already been published in *Traiperm & Staples (2016)*.

**Table 2** A comparison of the morphology and distributions of *Argyreia collinsiae* subsp. *collinsiae*, *A. collinsiae* subsp. *megabracteata*, *A. dokmaihom* and *A. versicolor* from Thailand.

| No. | Characters | A. collinsiae subsp. collinsiae | A. collinsiae subsp. megabracteata | A. dokmaihom | A. versicolor |
|---|---|---|---|---|---|
| 1 | Leaf blade shape | Orbicular | Orbicular or broadly ovate | Broadly ovate or ovate | Broadly ovate |
| 2 | Adaxial leaf surface | Puberulent | Puberulent | Strigose | Scabrous-strigose |
| 3 | Leaf base lobe length | 0.6-3.2 cm. | 1.2-4.1 cm | 2.9–3 cm | 0.7-1.1 cm |
| 4 | Inflorescence | Compound lax cymes | Compound lax cymes | Compound crowded cymes | Compound lax cymes |
| 5 | Floral odor | None | None | Fragrant | None |
| 6 | Bract color and shape | Green to reddish green, elliptic-oblong or narrowly lanceolate | Whitish green, ovate, obovate (rarely rhombic to orbicular) | Rose pink, ovate or rhombic | Whitish green, ovate |
| 7 | Persistence of bract at floral blooming stage | Caducous | Persistent | Persistent | Persistent |
| 8 | Bract size | 1.5–2.5(−3.5) ×0.7–1.1 cm | (2. 8-)3-5.6(−7.6) ×1.5–3.3(−4.5) cm | 2. 9× 3.1 cm | 3–4× 1.4–2 cm |
| 9 | Pedicel length | 0.5–1.5 cm | 0.1–0.7 cm | ca. 0.2 cm | 0.2–0.4 cm |
| 10 | Indumentum on mid-petaline bands | Glabrous | Glabrous | Glabrous | Sparsely pilose |
| 11 | Corolla limb color | Purple | Purple | White | Pale purple |
| 12 | Type of staminal glandular trichome | Short stalked | Short stalked | Short stalked | Short and long stalked |
| 13 | Type of staminal non-glandular trichome | Simple straight | Simple straight | Simple straight or branched | Simple straight |
| 14 | Distribution | Coastal areas and lower part of Chao Phraya Basin in Central region, as well as in Southwestern and Southeastern regions | Mekong Basin in Northeastern and Eastern regions | Kanchanaburi province, Southwestern region | Sa Kaeo province, Southeastern region |
| 15 | Estimated distribution area | 47,000 km$^2$ | 72,000 km$^2$ | Data deficient as only known from two localities | Less than 20 km$^2$ : only known from type locality |
| 16 | Elevation | 3–94 m asl | 132–220 m asl | 200–800 m asl | ca. 100 m asl |

### Taxonomic treatment
### Identification key for *A. collinsiae* and related taxa

The new couplets to place *A. collinsiae* should be inserted at couplet 10 in the *Flora of Thailand* (*Staples & Traiperm, 2010*:338), and some characters are modified from *Traiperm & Staples (2016)*.

1. Flowers pendulous; corolla limb purple or white, inside of tube white or pale brownish   2

1. Flowers facing upward (erect or ascending); corolla limb paler than the darker center (limb darker than tube in color variant of *A. variabilis*)                [return to couplet 11]

2. Corolla limb purple or pale purple, inside of tube white or pale brownish; bracts reddish green or green or whitish green; non-glandular trichomes simple and straight; inflorescence compound lax cymes                                                                             3

2. Corolla limb white, inside of tube pale brownish; bracts rose pink; non-glandular trichomes straight or branched; inflorescence compound crowded cymes               *A. dokmaihom*

3. Corolla limb purple, outside of midpetaline bands glabrous; adaxial leaf surface puberulent                                                                       *A. collinsiae*

3. Corolla limb pale purple, outside of midpetaline bands sparsely pilose; adaxial leaf surface scabrous-strigose                                                         *A. versicolor*

*Argyreia collinsiae*  (Craib) **Na Songkhla & Traiperm**, Thai Forest Bull., Bot. 33: 42. 2005 ('*collisae*') ≡ *Rivea collinsae* Craib in Bull. Misc. Inform. Kew 1916: 266. 1916 ≡ *Lettsomia collinsae* (Craib) Kerr in Bull. Misc. Inform. Kew 1941: 15. 1941. –Thailand. Sriracha, *A.F.G. Kerr 2149* (syntype: K barcodes K000830763! & K001081771!; isosyntypes: BK!, BM barcode BM000838989!, E barcode E00067043!); same locality, *Mrs D.J. Collins 53* (syntype: K barcode K001081770!; isosyntype: E barcode E00067042!).

Perennial woody twiners. Stems cylindrical, tips herbaceous, twining, whitish pubescent. Leaves orbicular broadly ovate, 4.4–21.1 by 4.3–17.2 cm, base cordate, apexe acute to acuminate, margins entire to undulate, chartaceous; indumentum both sides finely appressed puberulent; venation pinnate, grooved on adaxial sides slightly raised on the abaxial sides; petiole, 2–8 cm long, hirsute. Inflorescences axillary, pendulous, compound lax cymes, 3–many-flowered; peduncles 1–9 cm long, slightly pubescent; bracts elliptic-oblong or narrowly lanceolate, ovate, obovate or rarely rhombic to orbicular, 1.5–7.6 cm by 0.7–4.5 cm, green, reddish green or whitish green, margins entire to undulate, apexes acute to acuminate or long-attenuate, adaxial sides glabrous, abaxial sides slightly pilose, persistent or deciduous; pedicel 1–15 mm, slightly pubescent. Flowers diurnal, pendulous; sepals unequal, glabrous or rarely scanty hair, apex cucullate in the bud, outer two ovate or cordate or rarely orbicular, 1.0–1.8 by 0.8–1.6 cm, margins entire, apexes obtuse or acute, the third sepal unequal-sided, two inner sepals broadly obcordate or orbicular, 1–1.7 by 0.9–1.8 cm, margins entire, apex rounded or retuse, both sides glabrous; corolla campanulate, 3.8–6.6 cm long, thickly membranous, waxy, glabrous, limb entire, recurved, purplish, tube gibbous near middle, white outside and white or pale brownish inside, midpetaline bands glabrous. Stamens included, equal, whitish, filaments 1.4–2.3 cm long, adnate to corolla for 0.4–0.8 cm from the base, bases dilate; staminal trichome present, glandular and non-glandular trichomes distributed all around lower part of filaments or rarely glabrous; anthers oblong, 0.5–0.7 cm long. Pistils included; discs cupular, pentagonal, glabrous, subentire to undulate; ovaries sunken in nectary disc, globose, apex abruptly narrowing into style base, glabrous; styles 2.1–3.6 cm long, filamentous, glabrous; stigmas two-lobed. Fruit berry, globose to sub-globose, glabrous, reddish brown when ripe, glossy, enclosed by enlarged calyx. Seeds usually four, trigonous-rounded to hemisphere, dark brown, pubescent.

**Distribution:** Thailand, Cambodia, Vietnam and perhaps Laos.

**Habitat:** Coastal forest, mangrove, canal bank, roadside, in rice fields, limestone terrain or silty soil, dry deciduous forest, mixed evergreen-deciduous forest, canal bank, roadside, in rice fields.

**Phenology:** Flowering (August–) September–October (–November); Fruiting (October–) November–December.

**Notes:** In *Argyreia 'collinsae'* (Craib) Na Songkhla & Traiperm (*Khunwasi et al., 2005*), commemorating Mrs. Elian Emily Collins, the epithet is to be corrected '*collinsiae*' under Rec. 60.8(b) of the *Shenzhen Code* (*Turland et al., 2018*).

## Key to subspecies of *A. collinsiae* (Craib) Na Songkhla & Traiperm

1a. Bracts lanceolate or elliptic-oblong or narrowly lanceolate, 1.5–3.5 cm ×0.7–1.1cm, green to reddish green, caducous. Distribution: Southwestern, Central and Southeastern
$\qquad$*A. collinsiae* subsp. *collinsiae*

1b. Bracts ovate, obovate (rarely rhombic to orbicular), 2.8–7. 6× 1.5–4.5 cm, whitish green, persistent. Distribution: Northeastern and Eastern
$\qquad$*A. collinsiae* subsp. *megabracteata*

## 1a. Subsp. *collinsiae*

Inflorescences axillary, pendulous, compound lax cymes, 3–5 flowered; peduncles 1–2 cm long, slightly pubescent; bracts elliptic-oblong or narrowly lanceolate, 1.5–3.5 cm by 0.7–1.1 cm, green to reddish green, margins entire to undulate, apexes acute to long-attenuate, caducous.

**Distribution:** Thailand (Central, Southeastern and Southwestern).

**Habitat:** Coastal forest, mangrove, canal bank, roadside, rice fields, limestone terrain or silty soil.

**Phenology:** Flowering September–November, March; Fruiting September–January, March

**Conservation status:** Least Concern (LC) as it is a widespread species and not known to be at risk (*IUCN Standards and Petitions Committee, 2022*). The results were based on 20 collections of *A. collinsiae* subsp. *collinsiae* with an EOO greater than 63,586 km$^2$, and an AOO greater than 6,000 km$^2$.

**Additional specimens examined: Thailand.** Rayong: ca. km 233 along Rte. 3, 5 Nov 1985, *Staple and Promdej 247* (BKF); Mueang Rayong, Koh Samet, 27 Aug 2018, *P. Rattanakrajang et al. 146* (BKF); Mueang Rayong, Koh Samet, 27 Aug 2018, *P. Rattanakrajang et al. 147* (BKF); Prachaup Khiri Khan: Thap Sakae, Huey Yang Falls viewpoint, ca. 150 m alt., 3 Oct 2001, *Pooma et al. 3067* (BKF); Pran Buri, Khao Sam Roi Yot National Park, Tham Kaew, 18 Aug 2002, *Middleton et al. 1195* (BKF); Khao Chong Grackokk near Bang Saphan, 0-150 m alt., 18 Aug 1967, *Shimizu et al. T-7602* (BKF); 23 Aug 1982, *Shimizu et al. T-28725* (BKF); Saraburi: Hin Kong, ca. Km 101.5 along Rte. 33, 4 Nov 1985, *Staples and Wongprasert 218* (BKF); Mueang Saraburi, Kut Nok Plao, 14 Sep 2018, *Traiperm et al. 624* (BKF); Chonburi: Si Chang island, 5 m alt., 8 Nov 1992, *Maxwell 92-703* (CMUB, P); Si Chang island, Chudhaduth Palace, 20 Oct 2004, *Seelanan*

*et al. 420* (BCU); Khao Laem Ya–Mu Ko Samet National Park, 27 Nov 1988, *Khunwasi 32* (BCU); Sriracha, 30 Sep 1911, *A.F.G. Kerr 1942* (BM,K); Mueang Chonburi, Khao Sam Muk, 25 Aug 2018, *P. Rattanakrajang et al. 144* (BKF); Mueang Chonburi, Samet, Wat Bang Pang, 1 Sep 2018, *Y. Sirichamorn (2018) 10* (BKF); Kanchanaburi: ca. km. 17.5 along Rte. 323, 16 Nov 1985, *Staples and Wanthaniyakun 282* (BKF); Samut Prakan: Bang Pu, ancient city, 25 Aug 2018, *P. Rattanakrajang et al. 143* (BKF); Phetchaburi: Cha-am, Sirindhorn International Environmental Park, 30 Sep 2018, *P. Rattanakrajang et al. 150* (BKF); Nakhon Sawan: Phayuha Khiri, Sa Thale, 1 Oct 2021, *P. Srisombat et al. 40* (BKF); Suphan Buri: Don Chedi, Don Chedi, 6 Oct 2021, *P. Srisombat et al. 43* (BKF).

**1b. Subsp. *megabracteata* Traiperm & Srisombat**, subsp. nov.
**Type:** Thailand. Nakhon Ratchasima, Chok Chai, 14 Sep 2018, *P. Traiperm et al., 627* (holotype: BKF!, two sheets, numbered A & B; isotypes: BK!, K!, QBG!).

Inflorescences axillary, pendulous, compound lax cymes, many-flowered; peduncles 1–9 cm long, slightly pubescent; bracts ovate or obovate or rarely rhombic to orbicular, 2.8−7.6cm by 1.5–4.5 cm, whitish green, margins entire to undulate, apexes acute to acuminate, abaxial sides slightly pilose, adaxial sides glabrous, persistent. Figs. 2B, 2F, 2J, 2N, 2R, 2V, 5, 6.

**Distribution:** Thailand (Northeastern and Eastern).
**Habitat:** Dry deciduous forest, mixed evergreen-deciduous forest, canal bank, roadside, and rice fields
**Phenology:** Flowering September–November; Fruiting September–January
**Conservation status:** Least Concern (LC) as it is a widespread species and not known to be at risk (*IUCN Standards and Petitions Committee, 2022*). The results were based on 19 collections of *A. collinsiae* subsp. *megabracteata* with an EOO greater than 70,363 km$^2$ and an AOO greater than 6,000 km$^2$.
**Etymology:** The epithet '*megabracteata*' is derived from prominent bracts larger than the original mophotype.
**Notes:** *Argyreia collinsiae* subsp. *megabracteata* is similar to *A. collinsiae* subsp. *collinsiae* in having perennial woody twinning habit, orbicular leaves, and campanulate corollas, but differ from that species by having whitish green, ovate or obovate (rarely rhombic to orbicular) bracts, which are persistent (*versus* green to reddish green, elliptic-oblong or narrowly lanceolate bracts that usually deciduous), the distribution areas that are confined within Mekong Basin area in the Northeastern and Eastern, southerly and westerly constrained by mountain ranges (*versus* coastal areas and lower-elevated part in Southwestern, Central, and Southeastern regions).

This new subspecies is closely related to *A. dokmaihom* and *A. versicolor* in terms of leaf shape, hanging campanulate flower and bract shape, for details of differences see Table 2.

Given the ongoing and inconclusive discussion concerning the generic circumscription of *Ipomoea* and related genera (including *Argyreia*) (*Eserman et al., 2020*; *Eserman et al., 2024*; *Muñoz Rodríguez et al., 2019*; *Muñoz Rodríguez et al., 2023a*; *Muñoz Rodríguez et al., 2023b*) during the description of the new subspecies, the authors applied the widely accepted classification system of *Stefanović, Austin & Olmstead (2003)* throughout the work, where

*Argyreia* is recognized as a distinct genus, and the new subspecies was described under this generic name.

**Additional specimens examined: Thailand.** Surin: Thatum distr., ca. Km 36.1 along Rte. 214, Ban Nong Thatum, 27 Nov 1985, *Staples and Wongprasert 335* (BKF); Mueang Surin, Rajamangala University of Technology Esarn, Surin campus, 10 Sep 2018, *N. Chitchak & P. Traiperm 25* (BKF); Si Saket: en route from Si Saket city to Kantharalak about 10 km from Kantharalak, 130 m alt., 8 Oct 1984, *Murata et al. (T-49679)* (BKF); Loei: Phu Kradung, Ban Na Noi to Na Noi station (RS-2), 240-250 m alt., 26 Aug 1988, *Takahashi (T- 63160)* (BKF); Buriram: 21 Nov 1976, *Phengklai et al. 3340* (BKF); Nong Ki, Nong Ki, 10 Oct 2020, *P. Srisombat et al. 34* (BKF); Nong Khai: Tha Bo, 19 Aug 2018, *P. Hassa 17* (BKF); Maha Sarakham: Wapi Pathum, Khok Si Thong Lang, 28 Sep 2017, *N. Chitchak 14* (BKF); Wapi Pathum, Khok Si Thong Lang, 28 Sep 2017, *N. Chitchak & P. Traiperm 23* (BKF); Na Dun, 9 Sep 2018, *N. Chitchak & P. Traiperm 24* (BKF); Roi Et: Phon Thong, 11 Sep 2018, *N. Chitchak & P. Traiperm 26* (BKF); Nakhon Ratchasima: Chok Chai, Thung Arun, 14 Sep 2018, *P. Traiperm et al. 625* (BKF); Chok Chai, Thung Arun, 14 Sep 2018, *P. Traiperm et al. 626* (BKF); Chok Chai, Thung Arun, 24 Jan 2019, *P. Srisombat et al. 04* (BKF); Chok Chai, Thung Arun, Mai Don Ket village, 14 Sep 2018, *P. Traiperm et al. 627* (BKF); 24 Jan 2019, *P. Srisombat et al. 05* (BKF).

**An updated description from the Flora of Thailand** (*Staples & Traiperm, 2010*)
*Argyreia versicolor* **(Kerr) Staples & Traiperm**, Thai Forest Bull., Bot. 33: 43. 2005 ≡ *Lettsomia versicolor* Kerr in Bull. Misc. Inform. Kew 1941: 17. 1941 –**Holotype:** Thailand. [Sa Kaeo] "Krabin", [Watthana Nakhon] "Wattana", 27 Dec 1924, *A.F.G. Kerr 9786* (K barcode K000830750!; isotypes: BK barcode 257822!, BM barcode BM000898373!, P barcode P00391870!).

Woody twiner, all parts hispid to pilose with spreading hairs; stems terete, striate. Leaves ovate to broadly ovate, 5.6–10.8 by 3.7–9.5 cm, chartaceous, base cordate, apex acute-acuminate, adaxial side scabrous-strigose, abaxial densely strigose; lateral veins 6–10 per side; petiole stout- cylindrical, 1–4.7 cm, pubescent. Inflorescence axillary, many-flowered, compound lax cymes; peduncle stout, 1–1.7 cm; bracts numerous, whitish green, much larger than calyx, outer ovate, 3–4 by 1.4–2.3 cm, chartaceous, acuminate, outside sparsely appressed strigose, inner ones narrower; pedicels 2–4 mm. Flowers diurnal, pendulous; sepals unequal, outer two ovate or broadly ovate 1–1.3 cm by 0.8–1 cm, margins entire, apexes obtuse or acute, strigose abaxially, glabrous adaxially, the third sepal unequal-sided 1–1.3 cm by 0.9–1 cm, strigose to glabrous abaxially, glabrous adaxially, two inner sepals broadly obcordate or orbicular, 1–1.25 cm by 0.9–1 cm, margins entire, apexes rounded or retuse, strigose to glabrous abaxially, glabrous adaxially; corolla campanulate, 5–6.4 cm long, tube white, limb pale purple, outside sparsely, midpetaline bands pilose. Stamens included, filaments 1.5–2 cm, dilated and glandular at insertion, otherwise glabrous, anthers 0.5–0.6 cm. Pistils included, disc slightly undulate, ca. 0.7 mm high, ovary 2 mm high, glabrous. Fruit berry Figs. 2D, 2H, 2L, 2P, 2T, 2X, 7.
**Distribution:** Thailand (endemic)

**Habitat:** Scrub and disturbed areas (agricultural plantation and demonstrated dry deciduous forest)

**Phenology:** Flowering September–December; fruiting November–January

**Reproductive biology:** The species is self-incompatible, indicating a need of cross-pollination for fruit and seed set. *Xylocopa aestuans* Linnaeus, 1758 and *Xylocopa latipes* Drury, 1773 played role as putative pollinators (*Jirabanjongjit et al., 2024b*).

**Conservation status:** Only two individuals were rediscovered in 2018 in the area potentially around where the type specimens were collected. This area is now part of Burapha University, Sa Kaeo campus. One individual grows naturally within an agricultural plantation, and another was in a demonstrated dry deciduous forest: the latter one was torn down and found dead due to road construction a couple of years afterward. We classified this species as Critically Endangered (CR-D) because the plant has human activity as a plausible threat, according to IUCN conservation criteria (*IUCN Standards and Petitions Committee, 2022*).

**Additional specimens examined: Thailand.** Sa Kaeo, Burapha University, Sakaeo Campus, Watthana Nakhon, 15 Sep 2018, *Traiperm et al. 628*, 15 Sep 2018, *Traiperm et al. 629*, 7 Sep 2019, *Jirabanjongjit et al. 08*

## ACKNOWLEDGEMENTS

We would like to thank Dr. Chakkrapong Rattamanee, Dr. Yotsawate Sirichamorn, Dr. Tomoki Sando, Dr. Pacharaporn Sangyojarn, Mr. Phongsakorn Kochaiphat and Miss Piriya Hassa for their assistance in field collections. We thank Prof. Dr. Diogo B. Provete, Prof. Dr. Maria Teresa Buril, and an anonymous reviewer for their helpful comments on an earlier version of this manuscript.

### Funding

This research received financial support from Mahidol University (MU's Strategic Research Fund: 2023, awarded to Paweena Traiperm and Alyssa B Stewart). The funders had no role in study design, data collection and analysis, decision to publish, or preparation of the manuscript.

### Grant Disclosures

The following grant information was disclosed by the authors:
Mahidol University (MU's Strategic Research Fund: 2023).

### Competing Interests

The authors declare there are no competing interests.

### Author Contributions

- Poompat Srisombat conceived and designed the experiments, performed the experiments, analyzed the data, prepared figures and/or tables, authored or reviewed drafts of the article, and approved the final draft.

- Natthaphong Chitchak conceived and designed the experiments, performed the experiments, analyzed the data, prepared figures and/or tables, authored or reviewed drafts of the article, and approved the final draft.
- Pantamith Rattanakrajang conceived and designed the experiments, performed the experiments, analyzed the data, prepared figures and/or tables, authored or reviewed drafts of the article, and approved the final draft.
- Alyssa B. Stewart conceived and designed the experiments, performed the experiments, analyzed the data, authored or reviewed drafts of the article, and approved the final draft.
- Paweena Traiperm conceived and designed the experiments, performed the experiments, analyzed the data, authored or reviewed drafts of the article, and approved the final draft.

## Data Availability

The raw measurements are available in the Supplementary Files.

## New Species Registration

The following information was supplied regarding the registration of a newly described species:

*Argyreia collinsiae* (Craib) Na Songkhla & Traiperm subsp. *megabracteata* Traiperm & Srisombat, subsp. nov.

## Supplemental Information

Supplemental information for this article can be found online at http://dx.doi.org/10.7717/peerj.18294#supplemental-information.

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
