# Peer review of "The Argyreia collinsiae species complex (Convolvulaceae): phenetic analysis and geographic distribution reveal subspecies new to science"

_PeerJ, doi:10.7717/peerj.18294_

## Round 0.1 · original submission · Major Revisions

We have now received back the comments from two reviewers on your manuscript. Most of the comments from both of them point to the need of re-structuring the format and content of the paper. I strongly agree with them. From the title itself, this seems a catch-all manuscript that tries to embrace many - at times disparate - aspects of the study system, from ENMs for current and future climate, taxonomic work, and rediscovery of presumably lost species.
I don't think all of this can fit in a single paper and I'd like authors to deeply reflect on what's the core message they want to transmit. Is it the description of the new subspecies? Is it the modelling exercise? is it a monograph on A. versicolor? Then, stick with it and frame the paper around this single idea. This confusion is much evident in the Introduction, which brings a mix of subjects without a concise sequence of ideas. You can't have it all. For example, you don't need to have an ENM for future scenarios to support subspecies status.
Both reviewers also mention - to which I agree - that the sequence of information in the Results is backwards. Like R2 mentions, the results of statistical analysis should precede the description of subspecies. Both also point out the lack of clarity in terms of the number of specimens and populations used. This is crucial and should be better described in the Methods.
Like R1 mentions, I don't see the usefullness of having both the cluster and the Discriminant analysis if the goal is test the assignment of specimens/populations to a given species. Keep only the Discriminante Analysis, but I highly suggest you to use a DAPC instead here, which is available in the R package adegenet. With it, you don't need to conduct further tests, like ANOVA, to conduct hypothesis test. Why you have a subheading called "multivariate analysis' and a separate one called "statistical analysis"?
Move the identification keys to after the formal description of the subspecies, but merge the two into one.

L. 17-24: delete
Fig. 9 can be deleted, it doesn't make sense to do further tests on the dataset. See my suggestion on using DAPC.
Fig. 13 should be the first map
Fig. 11 and 12 can be merged.

Reviewer 1 ·

Basic reporting

The paper is a very thorough examination of the A. collinsiae complex. The references, background etc. are all in order. It is well presented, but the structure and some of the tone in the paper needs tackling to make it a more coherent and logical flow. The paper is a bit of a muddle, is it a revision of this complex or the description of a new variety. The methods seem to be a scatter-gun approach and don't really complement the results - for example, the authors state flower colour can be used for identification of each type which begs the question, what was the point of the phenetic analysis. The SDMs seem tacked on. If the paper is re-structured as a revision of the group with a consistent approach to each taxon, then that would read better.

Experimental design

This is fine, but unclear why the SDM is being used, seems like an MSc project using a range of different techniques polished up and put in a paper, not all of which make a coherent story. In addition, the phenetic analysis ends up being more or less moot as the authors state on l. 517 that the three taxa can be 'told apart by corolla color' - a character not used in the phenetic analysis. (And, line 517, why is 'careful observation' needed - color is pretty obvious, can be seen with the naked eye, unclear what is meant by that.) With the phenetic analysis, which are always 'chicken and egg', should you pre-assign each sample to a taxon, or just put them all in the analysis without any identification and see what the analysis reveals.

Line 122. 'Eighteen populations...' unclear what was sample, one plant per population, several, the whole populations? What was the sampling strategy - did the authors take many collections if the population was of many plants, were any populations of a single plant?
Line 124 Why didn't the authors use the herbarium data to map the potential locations? In Line 134 they use the herbarium data to plot the species on an SDM but didn't use those data to help direct sampling? Seems a strange descision to mention that but mention 'social media'.
Line 138 Are the conservation assessments global or national (or the plants endemic so they are both)?

Validity of the findings

Fine. But it's a very large sledgehammer to crack quite a small nut. To justify all the techniques used, it would be better if the paper was presented as a revision of the A. collinsiae group with descriptions of all taxa. It is unclear why there is no full description of A. collinsiae var. collinsiae, as presented, this seems back to front. There should be a full description of A. collinsiae, and the short description should be of the new variety. Why haven't the authors lectotypified all names? They are the specialists, so leaving that out is strange; it should be done.

The structure of the results seems to be back to front. From the analysis, the results should discuss which characters and groupings were found and then from that, the final part of the paper, should be the taxonomic presentation. Not sure what PeerJ format requires.

Line 235 'adaxial leaf surface texture scabrous-strigose' compare to line 351 'all parts hispid to pilose'. What is texture? Usually chartaceous or coriaceous, but is scabrous-strigose referring to hairs? Muddled terminology with regard to hairs.
Line 399 What is a 'packed inflorescence'? Probably better to describe as number of flowers per lenght of inflorescence axis. Packed usually reveals to vernation and how things are 'packed' before unfurling and maturing, is that what is meant?

Additional comments

The elephant in the room - the work of Wood, Scotland et al. who sink Argyreia into Ipomoea and do not consider the former an accepted genus. It is ironic that this is not tackled head on when compared to Line 487! Are there 'solid taxonomic boundaries' between Ipomoea and Argyreia?!?

In the Introduction the authors need to explicit describe and discuss A. collinsiae rather than talking about the ambiguity of the group.

Line 55 Add the authority at the first mention.
Line 55 'comprises three Argyreia species that have...'
Line 72. 'Several uncertain taxa' - it's just three. Several implies more, be specific.
Line 82 better as 'taxonomic status could be confirmed.'
Line 111 onwards. Is a discussion of IPNI part of the M+Ms? The authors do not write IPNI procedures, unlike why this is in there.
Line 462 best to have generic names in full at first mention in paragraphs?
Line 470 the regions in capitals?
Line 514 Should this be in the M+Ms?

·

Basic reporting

1. In general, the document is structured and written clearly. However, I suggest some minor adjustments:

* The authors use the term "original" or "original morphotype" to refer to the typical variety/subspecies of the species studied, throughout the text. I recommend it be replaced with "typical" subspecies.

Line 73 - comma before quote mark

Lines 145 to 154 - try to replace the verb “use/used” repeated many times

Line 508 - rewrite for clarity / its absence from where?

As I am not a native english speaker, I prefer to not judge the quality of the language.

2. I missed a better background on species complexes in Convolvulaceae. Some recent references that deal with similar problems were not cited by the authors, such as those dealing with Daustinia montana and Jacquemontia evolvuloides.

3. Regarding the structure of the manuscript, I consider more appropriate to present the taxonomic treatment after the statistical/modelling results and discussion, if the taxonomic decisions are made based on the results of the analysis.

4. Despite recognizing the merit of the research and its results, it is important to highlight that the authors did not present their hypotheses clearly, making it difficult to assess whether the methodology used was adequate to test them.

Experimental design

1. The research is original and fits the scope of PeerJ.

2. The main goal of the work is clear, and is relevant within discussions of the systematics of Convolvulaceae - as well as the flora as a whole, since species complexes are barriers to understanding biodiversity.

3. My main criticism of the work refers to the limitation of the data analyzed. How many specimens per population were included in the analyses? This is not clearly described in the methodology. Do the authors observe intrapopulation morphological variation? When dealing with species complexes, which often appear to be a continuum of variation, not only a large number of variables but also a large number of analyzed individuals is important. Another problem is that, from what I understood in my reading, the authors defined the morphotypes a priori and not a posteriori, which biases the data analysis.

4. Regarding the methods, please note the issues above mentioned.

Validity of the findings

1. The research has an important impact on the discussion of the validity of morphological characters and the delimitation of species in Convolvulaceae, in addition to being important for supporting conservation policies for the Thailand flora.

2. Underlying data have been provided and are robust.

3. The conclusions meets the results presented and the work main question. However, my second major criticism of the article is the designation of a new subspecies, when the results presented and the discussion would support the description of a new species (see that the authors debate the reproductive isolation of the treated morphotype).

Additional comments

As stated by De Queiroz in 2020, the taxonomic treatment of incompletely separated lineages is an old problem. Indeed, disagreements about whether to recognize certain groups as species versus subspecies have existed for centuries. I understand the difficult of taking taxonomic decisions when dealing with species complexes, with the risk of either overestimating or underestimating the number of species.
According to that author, the subspecies category should be reconsidered to encompass lineages that are not completely separate - and not as a taxonomic rank. I also understand that it was not the main goal of the authors to test the separation of lineages either by molecular or reproductive biology methods. However, when applying the morphological concept, based on several morphological traits that indicate a separation of the lineages, I would recommend that they improve their discussion on why not consider it as a new species. Besides that, it is important to make ir clear which species concept they are applying.

Other minor suggestions:

I suggest that a complete description of the species should be provided, with the entire range of variation considered by the authors, and then a diagnosis for both subspecies - the typical and the new one.

I also suggest that the notes bring a comparison with other species instead of comparing subspecies.

---

## Round 0.2 · Minor Revisions

Dear authors,

I have now received back the final comments from the same reviewers from the previous round. Like them I also think the paper is much improved and streamlined. I wouldn't have kept the ENM because it greatly diverges from the main goal, but I don't think this is necessarily a flaw of the paper. However, R2 still has some important questions about the very definition of OTUs, plus a couple of suggestions on the writing, that I think it's worth incorporating into the revision.

So, I'd like to invite authors to revise their paper accordingly.

Reviewer 1 ·

Basic reporting

The paper is much improved. Still extremely long and complex to describe a subspecies, and unclear what is the point of the SDM work.

Experimental design

Ok

Validity of the findings

Good

Additional comments

The paper is ok for publication if PeerJ wants to publish such things. It's a solid piece of work unpicking a species complex. Still no mention of generic issues with Ipomoea, but perhaps that doesn't even matter.

·

Basic reporting

1. Background: Line 62: when defining A. colinsae complex, the authors highlight the shape of bracts (line 54). If the new morphotype bears different bracts, in what set of characters is it similar to A. colinsae? Also clarify the similarity to A. versicolor. This is important to provide a good background of the problem to be treated in the manuscript.

Experimental design

1. Methods:

1.1. Explain how were the OTUs defined.
1.2. What is the character “lip color”? Is it regarding the corolla lobes?

Validity of the findings

No comment.

Additional comments

Some minor adjustments necessary on the Discussion:

Line 247 - solid taxonomic boundaries (among species?)
Line 355 - “which is probably an adaptation to improve visibility to birds” / should not be “which is probably adapted to dispersal by birds” instead?
Line 359 - add the author and year for “Macaca fascicularis” species

---

## Round 0.3 · accepted · Accept

Thank you for incorporating these last changes to the manuscript. I'm glad to recommend it for publication as is.